

# An improved primer set and amplification protocol with increased specificity and sensitivity targeting the *Symbiodinium* ITS2 region

Benjamin C.C. Hume[1], Maren Ziegler[1], Julie Poulain[2,3,4], Xavier Pochon[5,6], Sarah Romac[7], Emilie Boissin[8], Colomban de Vargas[7], Serge Planes[8], Patrick Wincker[2,3,4] and Christian R. Voolstra[1]

[1] Red Sea Research Center, Division of Biological and Environmental Sciences and Engineering (BESE), King Abdullah University of Science and Technology (KAUST), Thuwal, Saudi Arabia
[2] CEA–Institut de Biologie François Jacob, Genoscope, Evry, France
[3] CNRS, UMR 8030, Evry, France
[4] Université d'Evry, UMR 8030, Evry, France
[5] Coastal and Freshwater Group, Cawthron Institute, Nelson, New Zealand
[6] Institute of Marine Science, University of Auckland, Warkworth, New Zealand
[7] CNRS, UMR 7144, EPEP & Sorbonne Universités, UPMC Université Paris 06; Station Biologique de Roscoff, Roscoff, France
[8] PSL Research University: EPHE-UPVD-CNRS, USR 3278 CRIOBE, Université de Perpignan, Perpignan Cedex, France

Corresponding author
Christian R. Voolstra,
christian.voolstra@kaust.edu.sa

## ABSTRACT

The Internal Transcribed Spacer 2 (ITS2) rRNA gene is a commonly targeted genetic marker to assess diversity of *Symbiodinium*, a dinoflagellate genus of algal endosymbionts that is pervasively associated with marine invertebrates, and notably reef-building corals. Here we tested three commonly used ITS2 primer pairs (SYM_VAR_5.8S2/SYM_VAR_REV, ITSintfor2/ITSReverse, and ITS-DINO/ITS2Rev2) with regard to amplification specificity and sensitivity towards *Symbiodinium*, as well as sub-genera taxonomic bias. We tested these primers over a range of sample types including three coral species, coral surrounding water, reef surface water, and open ocean water to assess their suitability for use in large-scale next generation sequencing projects and to develop a standardised PCR protocol. We found the SYM_VAR_5.8S2/SYM_VAR_REV primers to perform superior to the other tested ITS2 primers. We therefore used this primer pair to develop a standardised PCR protocol. To do this, we tested the effect of PCR-to-PCR variation, annealing temperature, cycle number, and different polymerase systems on the PCR efficacy. The *Symbiodinium* ITS2 PCR protocol developed here delivers improved specificity and sensitivity towards *Symbiodinium* with apparent minimal sub-genera taxonomic bias across all sample types. In particular, the protocol's ability to amplify *Symbiodinium* from a range of environmental sources will facilitate the study of *Symbiodinium* populations across biomes.

## INTRODUCTION

Coral reefs sustain some of the highest levels of biodiversity on Earth and provide a range of services to communities totalling millions of people (*Moberg & Folke, 1999*; *Plaisance et al., 2011*). However, these ecosystems are being lost at an alarming rate (*Hughes et al., 2017*; *Norstrom et al., 2016*). This loss is primarily due to anthropogenic stressors degrading the scleractinian corals that build and support these reefs. Interestingly, susceptibility to these stressors amongst reef-building corals is often not homogenous (*Kemp et al., 2014*; *Pandolfi et al., 2011*; *Rowan et al., 1997*).

Coral resilience is determined not only by the animal genotype but also by the diversity of microbes that associate with the animal host (*Bourne, Morrow & Webster, 2016*; *Levin et al., 2017*; *Peixoto et al., 2017*; *Pogoreutz et al., 2017*; *Radecker et al., 2015*; *Torda et al., 2017*; *Ziegler et al., 2017c*). Of the microbial components that make up the coral holobiont—the consideration of the animal host and its associating microbes as a functional ecological unit—the algal symbionts of the genus *Symbiodinium* have received the most attention for their role in affording resilience to the coral host (*Hume et al., 2016*; *Thornhill et al., 2017*). The coral-*Symbiodinium* association is generally obligate with the algal symbiont providing towards the nutritional needs of the animal host, in exchange for a stable, beneficial environment (*Muscatine, 1990*). The efficacy and character of this association is determined by the genotype of alga (*Kemp et al., 2014*; *Rowan et al., 1997*; *Silverstein, Cunning & Baker, 2015*; *Thornhill et al., 2017*). Accordingly, the ability to resolve taxa within the genus *Symbiodinium* is essential to better understanding resilience of the coral holobiont and reef ecosystems as a whole.

Taxonomic resolution within *Symbiodinium* is primarily achieved genetically (*Wham, Ning & LaJeunesse, 2017*). To this end, a range of genetic markers and associated analytical approaches exist (*Pochon, Putnam & Gates, 2014*; *Thornhill et al., 2017*). Initially, genetic characterisations resolved *Symbiodinium* into broad taxonomic groupings referred to as clades A–I (*Pochon & Gates, 2010*). However, due to the significant genetic and phenotypic diversity found within these clades, contemporary genetic resolutions are conducted at a sub-clade level. For such analyses, the most commonly used marker is the Internal Transcribed Spacer 2 (ITS2) region of the nrDNA (*LaJeunesse, 2001*). This marker is multi-copy in nature meaning that a single *Symbiodinium* cell may contain 100s to 1,000s of copies of the ITS2 region (*Arif et al., 2014*; *LaJeunesse, 2002*). As such, 100s of distinct ITS2 sequences may be generated from a single genotype, referred to as intragenomic diversity. Given that corals may associate with multiple *Symbiodinium* genotypes, intergenomic diversity may also exist in samples. Disentangling these two diversities can complicate analyses, but ultimately, the intragenomic diversity of the ITS2 region has proven to be a rich source of information that has been effectively leveraged to improve resolutions within *Symbiodinium* (*Smith, Ketchum & Burt, 2017*; *Wham, Ning & LaJeunesse, 2017*).

*Symbiodinium* ITS2 intragenomic diversity was first employed taxonomically using denaturing gradient gel electrophoresis (DGGE) methodologies, enabling the separation of PCR amplicon sequence fragments according to melting temperature, a proxy for

sequence (*LaJeunesse, 2002*). Using such techniques, *Symbiodinium* taxa that shared the same most abundant ITS2 sequence in common, yet displayed significantly different functional phenotypes, were able to be resolved specifically according to differences in rarer sequences in their intragenomic diversity. Importantly, *Symbiodinium* taxa began to be resolved in part by specific sets of these intragenomic ITS2 sequences referred to as ITS2 types or profiles. For example, *Symbiodinium trenchii* and *Symbiodinium glynnii* have the D1 ITS2 sequence in common but can be differentiated according to different presence–absence combinations of the D4 and D6 sequences (*Wham, Ning & LaJeunesse, 2017*). Similarly, *Symbiodinium thermophilum*, a symbiont prevalent in corals living in the world's warmest reefs, can be differentiated from other C3 dominated symbionts, that have a global distribution, through the identification of a specific intragenomic sequence, the *S. thermo.*-indel (*Hume et al., 2015*).

More recently, studies have been taking advantage of the sequencing benefits afforded by next-generation sequencing (NGS) of PCR amplicons to elucidate *Symbiodinium* diversity using the ITS2 region. Multiple approaches to dealing with the sequence diversity have been taken, including, but not limited to, Operational Taxonomic Unit (OTU) clustering (*Arif et al., 2014*; *Cunning, Gates & Edmunds, 2017*; *Ziegler et al., 2017a*, *2017b*), searching for specific taxa-defining ITS2 sequences (*Boulotte et al., 2016*; *Hume et al., 2016*; *Ziegler et al., 2017a*), or looking for ITS2 profiles found in common between samples (*Smith, Ketchum & Burt, 2017*).

Given the ongoing surge in *Symbiodinium* diversity studies employing NGS approaches using the ITS2 marker, and the large-scale projects that rely on them, such as the Global Coral Microbiome Project (http://vegathurberlab.oregonstate.edu/global-coral-microbiome-project) and the *Tara* Pacific expedition (http://oceans.taraexpeditions.org/en/m/environment/ocean-climate/new-expedition-tara-pacific/), we sought to establish, and future-proof, a standardised and improved PCR protocol. Particularly, we assessed the specificity (preferential amplification of *Symbiodinium* DNA), sensitivity (ability to amplify *Symbiodinium* from *Symbiodinium*-rare environments), and relative sub-genera taxonomic bias (within *Symbiodinium* bias) of existing ITS2 primer pairs, as well as the effect of different polymerase systems. Notably, we tested amplification on a range of sample types including coral, coral surrounding water (CSW), reef surface water, and open ocean water due to increasing efforts to elucidate environmental reservoirs and assess free-living *Symbiodinium* (*Mordret et al., 2016*; *Nitschke, Davy & Ward, 2016*; *Pochon et al., 2010*; *Thornhill et al., 2017*).

## MATERIALS AND METHODS

### Sample collection, DNA isolation, ITS2 primer pairs, and experimental setup

To establish a robust ITS2 NGS-based amplification protocol for *Symbiodinium*, we assessed the effectiveness of three commonly used primer pairs (Table 1) on samples from different environments (Table 2). Further, we tested the robustness of the most effective primer pair, SYM_VAR, by documenting the effects of PCR-to-PCR variation,
**Table 1 Primer pairs tested.**

| Primer pair | Sequence 5′–3′[1] | Amplicon size bp | Protocol[2,3] | Cycles[1] | Publication |
|---|---|---|---|---|---|
| SYM_VAR_5.8S2[2] | GAATTGCAGAACTCCGTGAACC | ~234–266 | 98 °C for 2 min | 35 | *Hume et al. (2015)* |
| SYM_VAR_REV[2] | CGGGTTCWCTTGTYTGACTTCATGC | | 98 °C for 10 s, 56 °C for 30 s, 72 °C for 30 s 72 °C for 5 min | | *Hume et al. (2013)* |
| ITSIntFor2 | GAATTGCAGAACTCCGTG | ~258–290 | 95 °C for 2 min | 35 | *LaJeunesse (2002)* |
| ITS-Reverse | GGGATCCATATGCTTAAGTTCAGCGGGT | | 95 °C for 30 s, 52 °C for 30 s, 72 °C for 30 s 72 °C for 5 min | | *Coleman, Suarez & Goff (1994)* |
| ITS-DINO | GTGAATTGCAGAACTCCGTG | ~271–303 | 95 °C for 2 min | 35 | *Pochon et al. (2001)* |
| ITS2Rev2 | CCTCCGCTTACTTATATGCTT | | 95 °C for 45 s, 56 °C for 45 s, 72 °C for 45 s 72 °C for 5 min | | *Stat et al. (2009)* |

Notes:
[1] For this study, primer pairs were used with an Illumina MiSeq adapter sequence concatenated to the 5′ end (the same set of adapters were used for each primer pair). For example, the SYM_VAR primer pair was:
(MiSeq adapter) + SYM_VAR_5.8S2: 5′ (TCGTCGGCAGCGTCAGATGTGTATAAGAGACAG)GAATTGCAGAACTCCGTGAACC 3′
(MiSeq adapter) + SYM_VAR_REV: 5′ (GTCTCGTGGGCTCGGAGATGTGTATAAGAGACAG)CGGGTTCWCTTGTYTGACTTCATGC 3′
[2] Alternative number of cycles and annealing temperatures were tested during optimisation of the SYM_VAR primer set. See 'Materials and Methods' for details.
[3] If an alternative polymerase system to the Phusion HF PCR Master Mix is used, temperatures and times of cycling steps may need to be adjusted according to the polymerase manufacturer's guidelines.

annealing temperature, cycle number, and polymerase system used with regards to: specificity, sensitivity, relative sub-genera amplification, and which sequences were returned. Chimeric sequence formation was additionally assessed as a function of PCR cycles.

Sample collection: In total, 27 samples were used in our analyses from four different environments: coral tissue (CO), coral surrounding water (CSW), coral reef surface water (SRF-CO), and open ocean (SRF-OO). Samples were collected either as part of the *Tara* Oceans or *Tara* Pacific expeditions. Field permits were obtained as part of the *Tara* Oceans and *Tara* Pacific expeditions and the approving bodies were the respective governments and/or ministries for the environment of the countries where the samples were collected. The authors of this work are part of the *Tara* Oceans and *Tara* Pacific expeditions, and thus, participated in sample collection. *Tara* Expeditions are global scientific voyages aimed at probing morphological and molecular diversity, evolution, and ecology of marine plankton (viruses, bacteria, archaea, protists, and planktonic metazoans) in the photic layer of the world oceans to research how they are impacted by Earth's climate changes. *Tara* meteorological, oceanographic, biochemical, and plankton morphological data are archived in the comprehensive Sample Registry of the PANGAEA database (https://www.pangaea.de). The Sample Registry links physical samples to metadata about sampling and analysis methodology performed on each sample. Whilst the *Tara* Pacific registry is still under construction at the time of writing of this manuscript, the *Tara* Oceans registry can be accessed at https://doi.pangaea.de/10.1594/PANGAEA.842237.

Six CO samples from three genera, 2 *Porites lobata* (samples: CO-0000150 and CO-0000151), 2 *Pocillopora meandrina* (samples: CO-0000208 and CO-0000209), and 2 *Millepora platyphylla* (samples: CO-0000302 and CO-0000303), were collected using SCUBA off the coast of Panama in July 2016. Corals were stored in DNA/RNA Shield

**Table 2 Sample overview and PCR conditions.**

| Environment[1] | Primer pair | Cycle no. | Annealing temp | Samples[2] |
|---|---|---|---|---|
| **SYM_VAR_5.8S2/SYM_VAR_REV** | | | | |
| CO | SYM_VAR | 35 | 56 | 6 |
| SRF-CO | SYM_VAR | 35 | 56 | 1 |
| CSW | SYM_VAR | 35 | 56 | 2 |
| SRF-OO | SYM_VAR | 35 | 56 | 18 |
| **ITSintfor2/ITSReverse** | | | | |
| CO | ITSintfor | 35 | 56 | 6 |
| SRF-CO | ITSintfor | 35 | 56 | 1 |
| CSW | ITSintfor | 35 | 56 | 2 |
| SRF-OO | ITSintfor | 35 | 56 | 18 |
| **ITS-DINO/ITS2Rev2** | | | | |
| CO | ITS-DINO | 35 | 56 | 6 |
| SRF-CO | ITS-DINO | 35 | 56 | 1 |
| CSW | ITS-DINO | 35 | 56 | 2 |
| SRF-OO | ITS-DINO | 35 | 56 | 18 |
| **Additional samples analysed for SYM_VAR PCR optimisation testing** | | | | |
| CO | SYM_VAR | 25 | 56 | 6 |
| SRF-CO | SYM_VAR | 25 | 56 | 2 |
| CSW | SYM_VAR | 25 | 56 | 3 |
| SRF-OO | SYM_VAR | 25 | 56 | 2 |
| SRF-CO | SYM_VAR | 25 | 59 | 1 |
| CSW | SYM_VAR | 25 | 59 | 1 |
| SRF-OO | SYM_VAR | 25 | 59 | 2 |
| CO | Polymerase[3] | 30 | 56 | 6 |
| SRF-CO | Polymerase | 30 | 56 | 1 |
| CSW | Polymerase | 30 | 56 | 2 |
| SRF-OO | Polymerase | 30 | 56 | 2 |

Notes:
Samples analysed for primer pair comparisons and optimisations are denoted.
[1] Environment abbreviations: CO, coral; CSW, coral surrounding water; SRF-CO, surface water; SRF-OO, open ocean.
[2] Details of individual samples can be found in Table S1.
[3] Samples with the primer pair listed as Polymerase were amplified with the SYM_VAR primers, but using the Advantage 2 rather than the Phusion polymerase system.

(Zymo Research, Irvine, CA, USA). Two CSW and one SRF-CO samples were also collected off the coast of Panama in July 2016. CSW samples were collected (15–17 L) from water surrounding two *P. meandrina* coral colonies using a boat-mounted vacuum pump. SRF-CO samples were collected (18 L) directly from the surface of the reef. Water was pre-filtered through a 20 μm metallic sieve, and then vacuum filtered on a 3 μm polycarbonate filter of 142 mm diameter for 15 mins. After filtration, filters were immediately flash-frozen in liquid nitrogen.

In addition, six SRF-OO samples were collected off the coast of Panama and were processed as described above for the CSW and SRF-CO samples. Twelve additional samples were collected as part of the *Tara* Oceans expedition from a range of ocean basins

 

including the Pacific, North Atlantic, and Indian Ocean. These samples were filtered to a size fraction of 5–20 μm as described in section 6a of *Pesant et al. (2015)*.

DNA isolation: DNA was extracted using the ZR-Duet DNA/RNA MiniPrep Plus (Zymo Research, Irvine, CA, USA) kit for all coral samples. Further, DNA was extracted from all water samples as described in *Alberti et al. (2017)* for the 5–20 μm and 3–20 μm fractions. In brief, the protocol was based on simultaneous extraction of DNA and RNA by cryogenic grinding of cryopreserved membrane filters, followed by nucleic acid extraction with NucleoSpin RNA kits (Macherey-Nagel, Düren, Germany) combined with DNA Elution buffer kit (Macherey-Nagel, Düren, Germany).

ITS2 primer pairs: Three pairs of published ITS2 primer pairs specifically designed to amplify *Symbiodinium* DNA were tested. These were: SYM_VAR_5.8S2/SYM_VAR_REV (*Hume et al., 2013, 2015*), ITSintfor2/ITS-reverse (*Coleman, Suarez & Goff, 1994*; *LaJeunesse, 2002*), and ITS-DINO/ITS2Rev2 (*Pochon et al., 2001*; *Stat et al., 2009*). Primer details and amplification protocols are given in Table 1. For ease of comprehension, each primer pair will be referred to as the SYM_VAR, ITSintfor2, or ITS-DINO primer pair.

Experimental setup: Sequencing for this study was conducted in two consecutive efforts. In the first part, aimed at directly comparing the three primer pairs, each primer pair was used to amplify each of the 27 samples. In the second part, a subset of samples was amplified using the SYM_VAR primer pair with different PCR protocols. We tested the effect of number of cycles, annealing temperature, and polymerase system on selectivity, sensitivity, and stability to optimise the reaction. Details of the samples as well as the associated number of cycles and annealing temperatures used can be found in Table 2 and Table S1. All PCRs were carried out in 25 μL reactions with the following conditions: 12.5 μL Phusion High-Fidelity PCR Master Mix 2X (ThermoFisher Scientific, Waltham, MA, USA) or equivalent amount of Advantage 2 PCR system (Takara Bio, Nojihigashi, Japan), 1 μL of forward and reverse primer at a concentration of 10 μM, 0.75 μL of DMSO, 1 μL of genomic DNA at a concentration of between 5 and 10 ng, and 8.75 μL of ddH$_2$O with a GeneAmp PCR system 9700 (Perkin Elmer, Waltham, MA, USA).

## Library preparation, sequencing, and sequence quality control

For both sequencing efforts, all PCRs were conducted in triplicate before being pooled for sequencing. Pooled PCR products were purified using 0.8× AMPure XP beads (Beckmann Coulter Genomics, Brea, CA, USA), then aliquots of purified amplicons were run on an Agilent Bioanalyzer (Agilent Technologies, Santa Clara, CA, USA) using the DNA High Sensitivity LabChip kit to verify lengths and were quantified using a Qubit Fluorometer (ThermoFisher Scientific, Waltham, MA, USA).

One hundred ng aliquots of the amplicons were directly end-repaired, A-tailed, and ligated to Illumina adapters (Illumina, San Diego, CA, USA) on a Biomek FX Laboratory Automation Workstation (Beckman Coulter, Brea, CA, USA). Then, library amplification was performed using the KAPA Hifi HotStart NGS kit (KAPA Biosystems, Wilmington, USA). After library profile analysis by LabChip GX (Perkin Elmer, Waltham, MA, USA) and qPCR quantification using PicoGreen (ThermoFisher Scientific, Waltham, MA, USA)

in 96-well plates, each library was sequenced using 250 bp read length chemistry in a paired-end flow cell on a MiSeq instrument (Illumina, San Diego, CA, USA).

During sequencing, Illumina's real time analysis was run with default settings to remove clusters with the least reliable data. Output BCL files were converted to demultiplexed fastq files using Illumina's *bcl2fastq* package and in-house filtering was applied to reads that passed the Illumina quality filters (*Alberti et al., 2017*). Read pairs that mapped onto run quality control (QC) sequences (Enterobacteria phage PhiX174 genome) were removed using the *bowtie* package.

For each of the sequenced samples analysed in this study the same sequence QC was performed. Briefly, mothur_1.39.5 (*Schloss et al., 2009*) was used to create contigs from paired forward and reverse demultiplexed fastq.gz files using the *make.contigs* command. The *screen.seqs* command was then used (*maxambig = 0, maxhomop = 5*) to discard sequences putatively generated from sequencing errors. The *unique.seqs* command was used to create a non-redundant collection of sequences represented by a .name and .fasta file. The remainder of the QC was performed using both the .name and .fasta files produced. The *pcr.seqs* command (*pdiffs = 2, rdiffs = 2*) was used to trim the primer sequence regions from the returned sequences. In addition, this command was used to discard sequences in which the specific primer pairs could not be found— indicative of poor sequencing quality—allowing a deviation of no more than two nucleotides differences in either of the forward or reverse primer sequences. Next, *split.abund* was run (*cutoff = 2*) to discard sequences that were not found more than two times in each sequenced sample—again, to reduce incorporating sequences with sequencing errors. Sequences were once again made non-redundant using the *unique.seqs* command. Finally, the *summary.seqs* command was used to calculate the size distribution of the remaining sequences. Sequences below the 2.5% and above the 97.5% size percentiles were removed using the *screen.seqs minlength* and *maxlength* parameters. A summary of the number of sequences retained at three stages of the QC process (pre-QC, post-QC, and *Symbiodinium* only sequences) are given in Table S1 and Fig. S1.

## Sequence data analysis—primer pair comparisons

To assess which of the three primer pairs performed the best, each primer pair was assessed for specificity (i.e. preferential amplification of *Symbiodinium* DNA) and sensitivity (i.e. ability to amplify *Symbiodinium* from *Symbiodinium*-rare environments). Additionally, potential within-*Symbiodinium* taxonomic biases were assessed by comparing the proportions of sequences for each primer pair that belonged to each of the nine major lineages of *Symbiodinium*, clades A–I (*Pochon & Gates, 2010*). Our assumption was that if no or only small clade biases exist, the proportion of sequences from each clade should be similar between primers. Consequently, a large deviation in clade proportions by any one of the primer pairs was regarded as indicative of taxonomic bias.

The specific PCR conditions used for each primer pair are given in Table 1 with PCR reagents as detailed above. The sequences returned from each PCR were annotated using *blastn* and the NCBI 'nt' database according to their closest match to one of the following categories: *Symbiodinium*, dinoflagellate, stony coral, Hydrozoan, uncultured

eukaryote, other, or 'no match'. To do this, the .fasta file produced from the initial QC from each sample was run against NCBI's 'nt' database with the *max_target_seqs* argument set to 1 and an output format string of '*6 qseqid sseqid staxids sscinames sblastnames evalue*'. For sequences to be categorised as *Symbiodinium*, an e-value >50 was required (representing approximately a 100% coverage match at 80–85% nt identity) in addition to the closest match being of *Symbiodinium* origin. Additionally, closest match subject sequences were screened for two sequences in particular (JN406302 and JN406301), which are mis-annotated as *Symbiodinium* (highly divergent from any other *Symbiodinium* sequences) in the 'nt' database. Thus, query sequences matching these sequences were annotated as 'Unclutured eukaryote.' Notably, before controlling for this, 58% of all '*Symbiodinium*' sequences amplified by the ITSintfor2 and ITS-DINO primer pairs (0% for the SymVar primer pair) in the SRF-OO samples matched these sequences as their closest match (Fig. S2).

*blastn* was also used to associate *Symbiodinium* sequences to one of the nine clades. To do this, a .fasta file was created for each sample that contained all sequences that had previously been categorised as *Symbiodinium*. This file was used as an input for *blastn* with the *max_target_seqs* argument set to 1 and an output format string of '*6 qseqid sseqid evalue pident gcovs*'. A custom BLAST database was used that contained a single representative sequence for each of the nine clades (Data S1).

## Sequence data analysis—optimising the SYM_VAR PCR protocol

Having determined SYM_VAR as the most effective primer pair for the amplification of the *Symbiodinium* ITS2 region, we undertook a second sequencing effort to optimise the PCR protocol. Specifically, we investigated: (1) how consistent the PCR was (i.e. PCR-to-PCR effect) by conducting PCRs with the exact same reaction conditions but sequenced on different sequencing runs, (2) whether increasing the annealing temperature could increase sensitivity in non-coral sample environments, and (3) whether a decrease in the number of PCR cycles would still allow for a satisfactory amplification of *Symbiodinium* whilst decreasing chimeric sequence return.

PCR conditions were the same as for the SYM_VAR_5.8S2/SYM_VAR_REV primer pair (Table 1), but with a 30 s denaturing step, following the manufacturer's recommendation (Phusion High Fidelity PCR; ThermoFisher Scientific, Waltham, MA, USA). Taxonomic identities and clade breakdown (*Symbiodinium* sequences only) of sequences returned from each sample (annealing temperature and cycle number comparisons only) were determined in the same way as for the initial primer pair comparisons described above.

As a final optimisation of the SYM_VAR amplification protocol, we tested whether the polymerase enzyme system used could affect the outcome of the PCR amplification. For 11 samples (Table 2), we contrasted amplification with Phusion High-Fidelity PCR Master Mix (ThermoFisher Scientific, Waltham, MA, USA) and the Advantage 2 PCR Enzyme System (Takara Bio, Tokyo, Japan). PCR amplification with the Advantage 2 PCR Enzyme System were conducted as shown in Table 1, but with 30 cycles.

To further assess the potential effect of PCR protocol modifications, we additionally assessed whether PCR-to-PCR variation, or any of the protocol optimisation modifications would affect which sequences were amplified. To assess this, the top 20 most abundant sequences (by relative abundance) were identified and plotted by decreasing abundance for pairings of sequenced samples for which PCR conditions were identical, or only differed in one of the investigated parameters (i.e. cycle number, annealing temperature, or polymerase system). This analysis was conducted for two samples, AW-0000035 and IW-0000015 and only for *Symbiodinium* sequences contained in each. It should finally be noted that although we did not test for it here, variability between different PCR cyclers (e.g. in ramping rate and accuracy of temperature binning) also has the potential to cause variation in a given PCR.

An online version of the final optimised PCR protocol is available through the Doi: 10.17504/protocols.io.n8edhte.

### Design of the SYM_VAR primer pair

The SYM_VAR primer pair were designed to minimise clade bias and coral-host complementarity. Originally, the reverse primer of the SYM_VAR primer pair that binds in the 28S region of the rDNA, SYM_VAR_REV, was designed to be complimentary to the forward primer SYM_VAR_FWD that binds in the 18S (*Hume et al., 2013*). This primer pair was designed to amplify both ITS regions as well as the 5.8S region of the rDNA. These primers were designed with the incorporation of degenerate bases based upon a multiple sequence alignment containing representatives of all *Symbiodinium* clades so that, where possible, the primer sequences had an equal number of nucleotide mis-matches to each of the clades. Later, to be able to amplify only the ITS2 region of the rDNA, the SYM_VAR_5.8S2 primer was designed (*Hume et al., 2015*). Compatibility to the SYM_VAR_REV primer and a minimisation of clade bias were maintained whilst complementarity to coral host ITS2 rDNA sequences was also taken into account by including representative sequences from *P. lobata* and *P. lutea* (ITS2 sequences from which had been amplified using alternative primers) to the multiple sequence alignment (*Hume, 2013*). It should be noted that this primer pair was successfully used to recover *Symbiodinium* DNA from *Acropora downingi*, *Cyphastrea microphthalma*, *Favia pallida*, *Platygyra daedalea*, *Porites harrisoni*, and *P. lutea* in *Hume et al. (2015)*.

### Sampling permits

The following permits were required for the collection of the samples analysed as part of this study: Panama scientific sampling: SE/AP-18-16; Panama CITES: SEX/A-72-16, no. 05567; Portugal scientific sampling: 080522/2009/Proc.o E.17.b; Maldives scientific sampling: FA-D2/33/2010/02; Chile scientific sampling: 13270/24/74.

## RESULTS

### Primer pair comparisons for amplification of *Symbiodinium* ITS2

In the CO samples, the SYM_VAR primer pair exclusively amplified *Symbiodinium* DNA. For the ITSintfor2 and ITS-DINO primer pairs *Symbiodinium* sequences represented

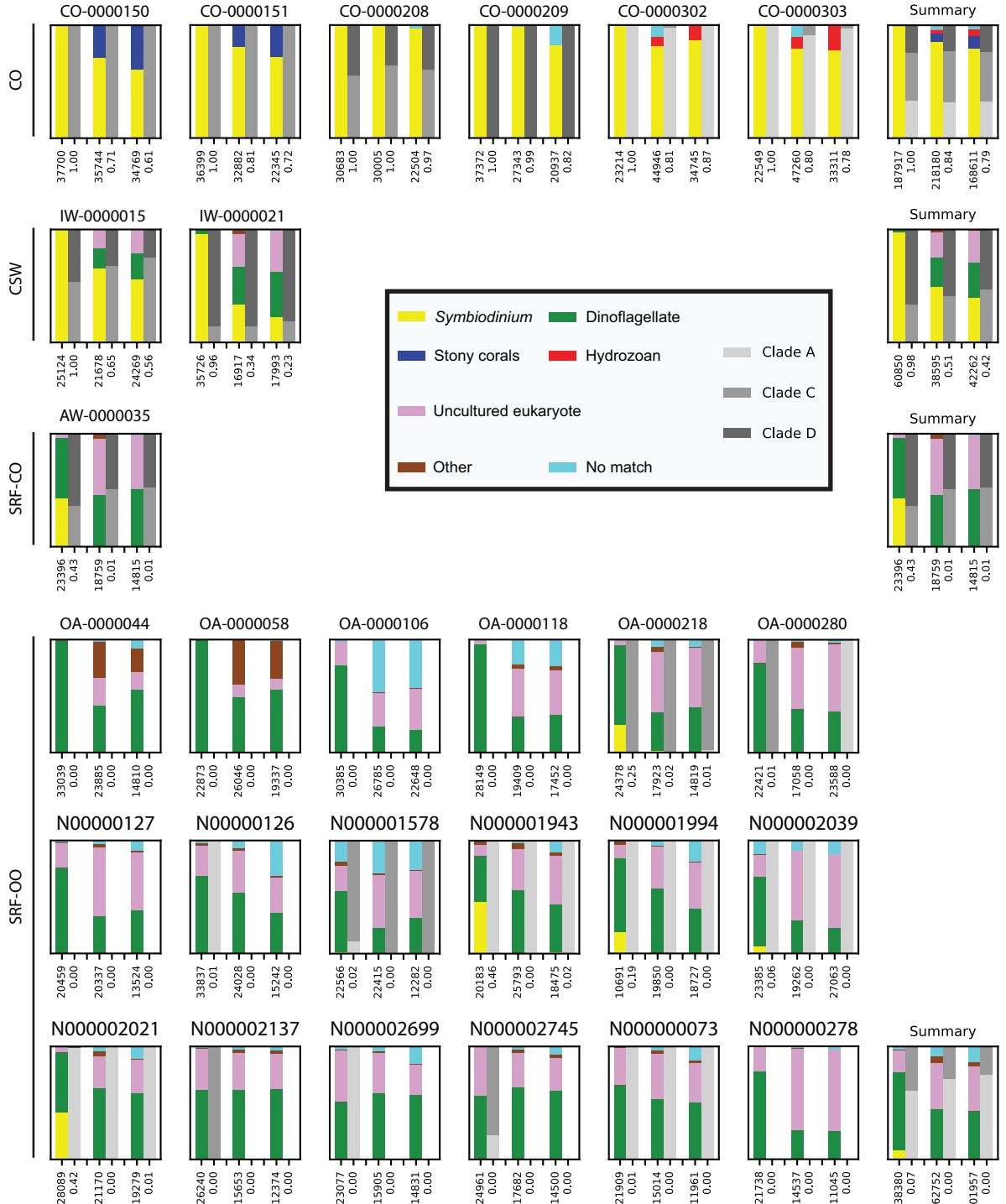

**Figure 1 Comparison and taxonomic composition of sequences returned using three *Symbiodinium* ITS2 primer pairs.** Each plot represents one sample and contains three pairs of stacked columns, one for each primer pair in the order of: SYM_VAR, ITSintfor2, ITS-DINO. For each pair of stacked columns, the coloured bar denotes the taxonomic breakdown of all returned sequences into seven categories, and the greyscale bar denotes the sub-genus (clade) distribution for the subset of sequences classified as *Symbiodinium* (see colour legend). Plots are annotated with the number of sequences returned after QC and the proportion of those sequences that were annotated as *Symbiodinium* underneath the coloured and grayscale bars, respectively. For each environment type a summary plot is given, showing the average taxonomic breakdowns, total post-QC sequences returned, and average proportion of *Symbiodinium* sequences.

84% (SD 11%) and 79% (SD 12%) of total reads, respectively. Remaining sequences associated with Scleractinia, Hydrozoa, or resulted in no BLAST match (Fig. 1).

In the CSW water samples, the SYM_VAR primer pair maintained a high specificity for *Symbiodinium* (98%; SD 3%). In contrast, the alternative pairs produced 51% (SD 23%) and 42% (SD 24%) of *Symbiodinium* reads, respectively, with the remaining sequences characterised as dinoflagellate or falling into the uncultured eukaryote category (Fig. 1).

In the SRF-CO sample, *Symbiodinium* amplifications were considerably lower compared to the CO and CSW samples. SYM_VAR amplified 43% of *Symbiodinium* reads compared to 1% in both alternative primer pairs. The remaining sequences in the SYM_VAR pair were primarily of dinoflagellate origin. In the other two primer pairs, although a large proportion were also dinoflagellate in origin, sequences primarily fell into the uncultured eukaryote category (Fig. 1).

In the SRF-OO samples, the ITSintfor2 and ITS-DINO primer pairs were more similar in their amplifications compared to the SYM_VAR pair. In contrast to the other primer pairs, SYM_VAR amplified less 'uncultured eukaryotic,' 'other,' and 'no match' sequences, and more dinoflagellate and *Symbiodinium* sequences. Specifically, amplification of *Symbiodinium* was achieved in 12, 7, and 8 of the SRF-OO samples using the SYM_VAR, ITSintfor2, and ITS-DINO primer pairs, respectively. All of the five libraries that achieved a *Symbiodinium* amplification above 5% of total reads were amplified with the SYM_VAR primer pair. In total, the SYM_VAR primer pair achieved a *Symbiodinium* amplification of 7% (SD 15%) compared to <0.5% (SDs < 0.5%) for the alternative primer pairs (Fig. 1).

Across all samples in which all three primer pairs amplified >5% of *Symbiodinium* reads, the proportions of sub-genus taxa amplified (*Symbiodinium* clades) were comparable (Fig. 1).

## Establishment of an optimal PCR protocol using the SYM_VAR primer pair

First, we assessed whether the same *Symbiodinium* ITS2 sequences were returned from PCRs performed on the same samples, with identical reaction conditions, but sequenced on different sequencing runs. For the two samples tested, the 20 most abundant sequences were present in both PCRs with relative abundances of each sequence being comparable (Fig. 2).

Next, we increased the annealing temperature from 56 to 59 °C to see whether a gain in *Symbiodinium* selectivity, and therefore sensitivity, could be achieved without yielding sub-genus taxonomic bias. In the CSW and SRF-CO samples, increasing annealing temperature increased the proportion of returned *Symbiodinium* sequences from 95% to 98% and from 28% to 47%, respectively. However, in both samples a small increase in the proportion of clade D versus clade C was observed, from 45 to 47% and from 57% to 66%, respectively. In the two SRF-OO samples tested, no *Symbiodinium* amplification occurred at either 56 or 59 °C (Fig. 3). The 20 most abundant sequences were found in both libraries for the CSW sample tested. However, two of the 20 sequences were

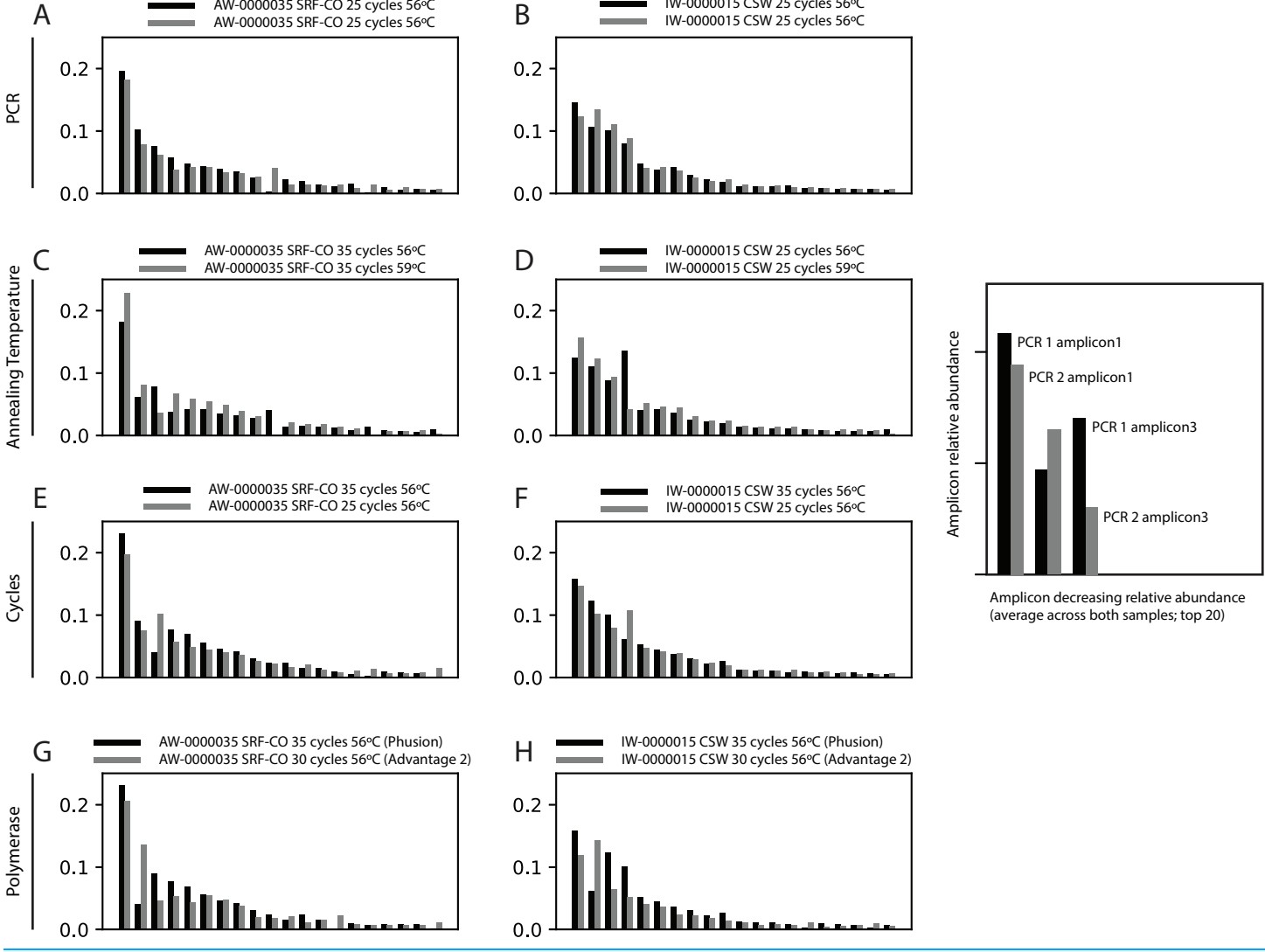

**Figure 2 Robustness of PCR protocol using the SYM_VAR ITS2 primer pair.** Each subplot represents a pairwise comparison of PCRs from the same sample. The sample, environment type, and PCR conditions are noted above each subplot. Depending on which pair of PCRs are being compared, these subplots graphically represent effects due to (A, B) PCR-to-PCR variation, (C, D) annealing temperature, (E, F) number of PCR cycles, or (G, H) polymerase system used. For each subplot, the relative presence of the 20 most abundant amplicons (calculated across both PCRs) is plotted for each PCR, denoted as black and grey bars. For example, (B) represents a pairwise comparison of PCRs run using the same annealing temperature and number of cycles on sample AW-0000035. It is therefore testing the PCR-to-PCR effect.

missing from one of the libraries in the SRF-CO sample tested (the 10th and 16th most abundant sequences; Fig. 2).

Additionally, we tested whether *Symbiodinium* could be sufficiently amplified using a decreased number of PCR cycles to reduce chimeric sequence formation or taxonomic bias. To accomplish this, we compared the above-used protocol to one with only 25 cycles. In the CSW and SRF-CO samples there was a greater proportion of *Symbiodinium* amplified at 35 compared to 25 cycles (~5% greater in the CSW samples and 15% in the SRF-CO samples). However, in both of the SRF-OO samples no *Symbiodinium*

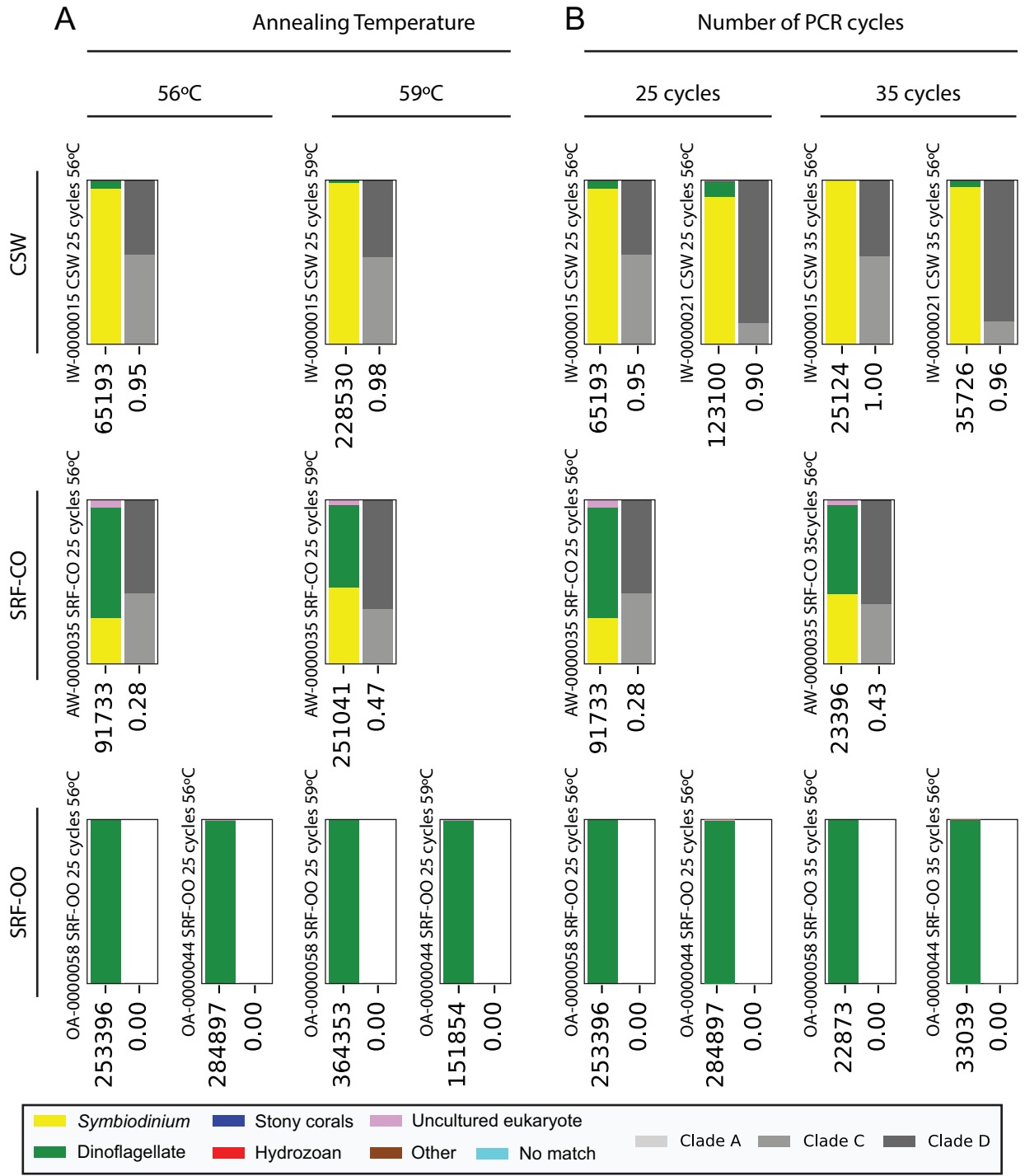

**Figure 3 Effect of annealing temperature and number of PCR cycles on sequences returned using the SYM_VAR ITS2 primer pair.** Two annealing temperatures (A; 56 and 59 °C; columns) and two cycle numbers (B; 25 and 35 cycles; columns) were tested on samples from three different environment types (rows). Each plot represents a different pooled PCR amplicon with sample name, sample type, number of cycles, and annealing temperature noted. For each plot, coloured bars denote the taxonomic breakdown of all returned sequences into seven categories. The greyscale bars denote the sub-genus (clade) distribution for the subset of sequences classified as *Symbiodinium*. Plots are annotated with the number of sequences returned after QC and the proportion of those sequences that were annotated as *Symbiodinium* underneath the coloured and greyscale bars, respectively. *Symbiodinium*-derived sequences were further subcategorized according to their clade identity (greyscale stacked bars).

**Table 3 Effect of number of PCR cycles on chimeric amplicon formation.**

| Sample | Chimeric seqs | Total seqs | Chim/tot. | Chimeric seqs | Total seqs | Chim/tot. |
|---|---|---|---|---|---|---|
| | Sequencing run 1 (25 cycles 56 °C) | | | Sequencing run 2 (35 cycles 56 °C) | | |
| AW-0000035 | 0 | 91733 | 0.00 | 35 | 23396 | 0.00 |
| IW-0000015 | 434 | 65193 | 0.01 | 626 | 25124 | 0.02 |
| IW-0000021 | 23 | 123100 | 0.00 | 481 | 35726 | 0.01 |
| CO-0000150 | 0 | 99626 | 0.00 | 0 | 37700 | 0.00 |
| CO-0000151 | 0 | 102149 | 0.00 | 0 | 36399 | 0.00 |
| CO-0000208 | 718 | 92663 | 0.01 | 758 | 30683 | 0.02 |
| CO-0000209 | 220 | 106830 | 0.00 | 0 | 37372 | 0.00 |
| CO-0000302 | 0 | 99784 | 0.00 | 0 | 22549 | 0.00 |
| CO-0000303 | 20 | 128780 | 0.00 | 0 | 23214 | 0.00 |

amplification was seen at either cycle conditions. Only non-*Symbiodinium*, dinoflagellate taxa sequences were amplified. Of the nine samples considered in the chimeric analysis (the three CSW and SRF-CO samples considered above, as well as six CO samples) greater number of chimeric sequences were putatively identified at 35 cycles versus 25 cycles in three of them with an increase of 1% in each case (Table 3). The proportions of sub-genus taxa amplified (*Symbiodinium* clades) were comparable (Fig. 3) and all but one (the 20th) of the most abundant sequences were amplified in common (Fig. 2).

Last, we also tested to see whether there would be any effect of using different polymerase systems. In the CO and SRF-OO samples tested, no difference in proportion of returned *Symbiodinium* sequences was observed between polymerase systems used (100% and 0% amplification for all libraries, respectively). However, in the CSW and SRF-CO samples we observed a relative decrease of *Symbiodinium* reads in the libraries that used the Advantage 2 system (Fig. 4). Despite, this relative decrease, the polymerase system used had minimal effect on which *Symbiodinium* sequences were returned (top 20 most abundant; Fig. 2).

## DISCUSSION

Next-generation sequencing approaches are quickly superseding their traditional gel-based counterparts due to advances in sequencing power and ease of handling (*Arif et al., 2014*; *Thomas et al., 2014*). In particular, increases in sequencing depths by several orders of magnitude have enabled the examination of low-abundance ITS2 sequences for the first time. The illumination of these rare-sequences holds the potential to characterising the rare *Symbiodinium* biosphere *in hospite* and in free-living contexts (*Boulotte et al., 2016*; *Mordret et al., 2016*; *Ziegler et al., 2017b*). Also, the incorporation of rarer ITS2 sequences into phylogenetic analyses has the potential to vastly improve the taxonomic resolving power of the ITS2 marker (*Smith, Ketchum & Burt, 2017*).

As rarer ITS2 sequence variants become more integral to the biological inferences made (*Boulotte et al., 2016*; *Ziegler et al., 2017b*), the effect of differences in ITS2 amplification strategies will become more significant. For example, small changes in amplification strategy may have a large effect on the rarer sequence variants returned. Despite this potential sensitivity to amplification variation, NGS studies published to date have used a

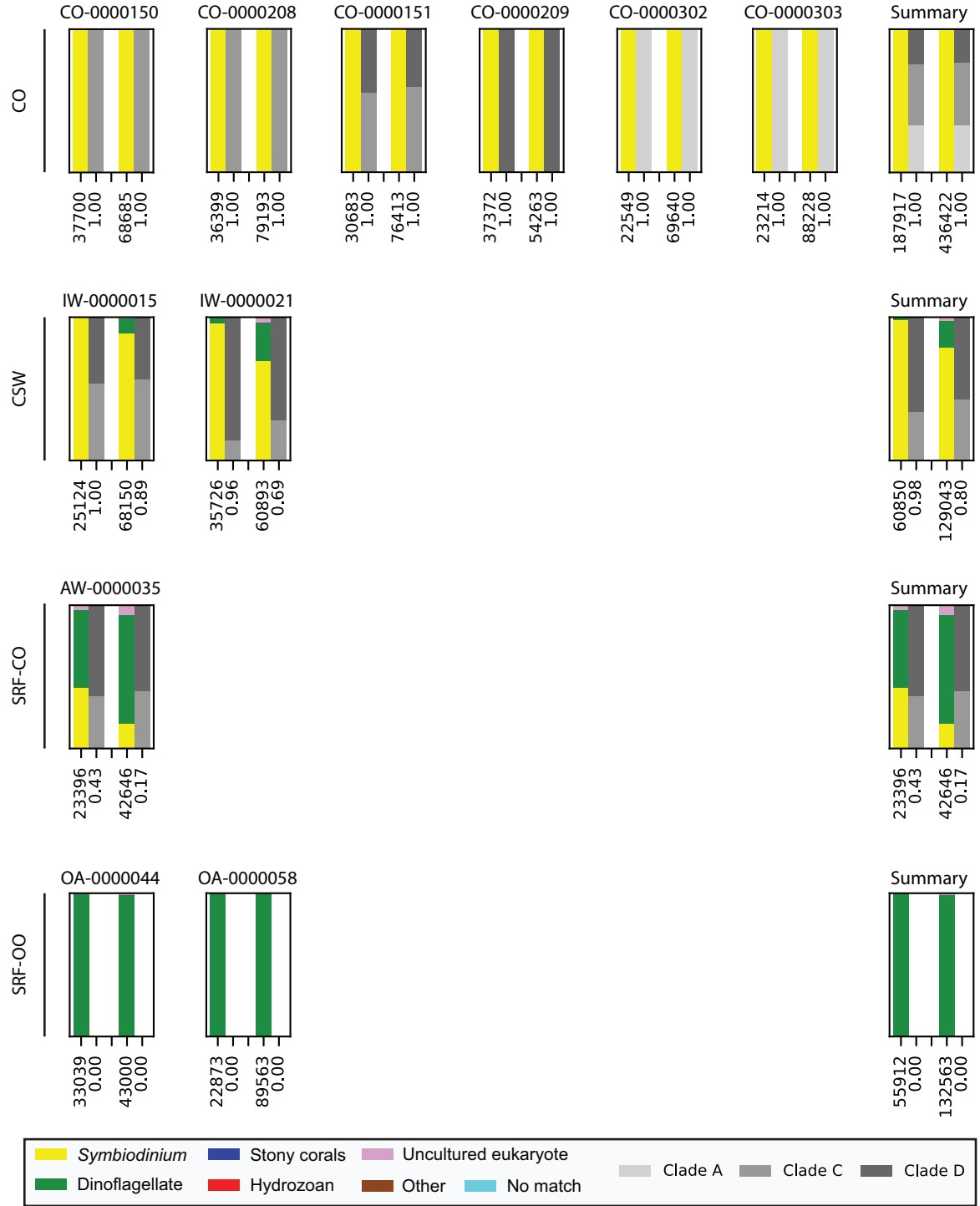

**Figure 4 Effect of polymerase system on sequences returned using the SYM_VAR ITS2 primer pair.** Each plot represents one sample and contains two pairs of stacked columns, one for each polymerase with Phusion (ThermoFisher Scientific, Waltham, MA, USA) on the left, and Advantage 2 (Takara Bio, Tokyo, Japan) on the right. Coloured bars denote the taxonomic breakdown of all returned sequences into seven categories. The greyscale bars denote the sub-genus (clade) distribution for the subset of sequences classified as *Symbiodinium*. Plots are annotated with the number of sequences returned after QC and the proportion of those sequences that were annotated as *Symbiodinium* underneath the coloured and greyscale bars, respectively. Plots are ordered by environment (rows; CO, coral; CSW, coral surrounding water; SRF-CO, surface water; SRF-OO, open ocean). A summary plot is provided for each environment type.
range of different primer pairs developed for use in less-sensitive analytical contexts, without first assessing the effect of primer choice. For example, ineffective amplifications have the potential to compromise our ability to infer taxonomic identities, and inter-primer pair variation could hinder cross-study comparisons. In particular, since several large-scale projects including the Global Coral Microbiome Project and the *Tara* Pacific expedition, in which thousands of samples are being analysed for *Symbiodinium* ITS2 diversity, many of which represent *Symbiodinium*-rare environments, it is important to standardise and test-proof the primer pair and PCR protocol employed. Here, we compared three commonly used *Symbiodinium* ITS2 primer pairs, and show that the SYM_VAR primer pair outperforms the two alternative pairs in specificity and sensitivity whilst maintaining an apparent minimal taxonomic bias.

Our data show that primer pair had a considerable effect on amplification efficacy of the *Symbiodinium* ITS2 gene, especially in *Symbiodinium*-rare environments. In the SRF-OO and SRF-CO samples, the SYM_VAR primer pair significantly outperformed the alternative primers amplifying more *Symbiodinium* reads in every sample tested (Fig. 1). Only SYM_VAR were able to amplify *Symbiodinium* in the majority of SRF-OO samples with proportions as high as 46%. In contrast, the maximum amplification of the alternative primers was 2% with both primer pairs returning averages of <1% from successful amplifications in less than half the samples. In more *Symbiodinium*-abundant samples, such as the CO or CSW samples, the use of the SYM_VAR primers enabled a significant gain in the proportion of *Symbiodinium* sequences returned, once again amplifying more *Symbiodinium* in every sample tested (Fig. 1). In practical terms, this translates to needing a shallower sequencing effort per sample to return the same amount of *Symbiodinium* information, thus enabling more samples to be sequenced per unit cost.

As part of optimising the SYM_VAR amplification protocol we tested whether increasing annealing temperature could increase the primers' taxonomic specificity for *Symbiodinium*. An increase in annealing temperature from 56 to 59 °C resulted in a higher proportion of *Symbiodinium* being amplified (Fig. 3). However, this was accompanied by a minor increase in the proportion of clade D *Symbiodinium* amplified. Given that any change in clade proportion resulting from an increased annealing temperature is most likely due to preferential amplification of certain clades over others, this increase may be symptomatic of a weak clade bias. Considered alongside the fact that sufficient *Symbiodinium* sequences were amplified to enable a robust analysis in the CO, CSW, and SRF-CO samples at 56 °C, and that increasing annealing temperature did not enable the amplification of *Symbiodinium* DNA in the SRF-OO samples, we recommend the 56 °C annealing temperature to preference minimising clade bias at the expense of additional *Symbiodinium* amplification. Notably, although every effort has been made to design the SYM_VAR primer pair in a manner that minimises *Symbiodinium* clade bias, further testing for clade bias in particular when working with *Symbiodinium* clades not recovered in this study may be needed to unequivocally confirm minimal bias. To this end, amplification, sequencing, and analysis of pre-determined, artificially mixed communities of *Symbiodinium* may offer an effective way to test for clade bias.

We also tested whether a reduced number of PCR cycles could still effectively amplify *Symbiodinium* DNA and whether the cycle number would affect chimeric sequence production. Although no *Symbiodinium* amplification was achieved in the 2 SRF-OO samples tested, a greater amplification of *Symbiodinium* at 35 rather than 25 cycles in the CSW and SRF-CO samples (Fig. 3), accompanied by only minimal reduction in chimeric sequence production (Fig. 2) at the lower cycles, advocates the use of the higher cycles in the standardised protocol. Notably, any cycles required for the preparation of NGS libraries should be included in the number of total cycles. For example, if adapter and index attachment require a total of 7 cycles, then 28 cycles should be used for the initial PCR. Otherwise 35 cycles should be used for non-NGS applications.

Finally, we investigated whether the polymerase system used could modify the specificity of the SYM_VAR pair. Importantly, our results showed that the use of the Advantage 2 rather than the Phusion system led to a decreased specificity for *Symbiodinium* DNA when used in conjunction with the SYM_VAR pair (Fig. 4). Whilst it is not our objective to recommend the use of one polymerase system over any other, and it is likely unrealistic to expect research groups to diverge from polymerase systems established in their labs, our results do highlight that the polymerase system used can have an effect on specificity and should therefore be taken into consideration during PCR optimisations. Reassuringly, the profile of ITS2 sequences returned from each of the polymerase systems were comparable (Fig. 2). This would suggest that although polymerase system may affect the efficiency of amplification of *Symbiodinium*, it will have no effect on inferences made from data.

Alongside considerations of *Symbiodinium* amplification efficacy (i.e. whether *Symbiodinium* can be amplified), it is also important to assess amplifications in terms of what taxonomic inferences they may offer. Due to its multicopy nature and the circumstance that more than one *Symbiodinium* taxa can be associated with any given host/environment, the level of taxonomic resolution offered by the ITS2 genetic marker within *Symbiodinium* is dependent on the depth of sequencing. As the returned number of *Symbiodinium* sequences decreases, so does the power to resolve. When 100s to 1000s of sequences (per clade) are returned, ITS2 type profiles (sets of intragenomic sequences diagnostic of *Symbiodinium* taxa) can be assessed in detail (*Smith, Ketchum & Burt, 2017*). Analyses that make use of intragenomic variance in such a way are very recent or still in development. Thus, species descriptions have yet to make use of NGS-derived ITS2 type profiles, but their identification has the ability to resolve at, and beyond, the species level within *Symbiodinium* (*Hume et al., 2015*; *LaJeunesse et al., 2014*; *Smith, Ketchum & Burt, 2017*). Whilst all of the primer pairs were able to return such sequencing depth in the CO and CSW samples, for the SRF-CO and SRF-OO samples this depth was only achieved by the SYM_VAR primer pair (and not consistently across all samples). With this resolution, correlations between *in hospite* and free-living *Symbiodinium* populations can be addressed with a high degree of certainty. Notably, the level of sequencing depth becomes rather an issue of sequencing effort. Hence, differences in sequencing efforts for different environments will need to be taken into account if comparisons are planned to be made between different environments at the same resolution.

## CONCLUSION

Given the ongoing popularity and increasing interest in phylotyping *Symbiodinium* in host as well as non-host environments employing the ITS2 region, we set out to develop a standardised *Symbiodinium* ITS2 amplification protocol, in particular for the use of NGS approaches. We show that the SYM_VAR_5.8S2/SYM_VAR_REV primer pair represent a superior primer pair able to amplify a greater proportion of *Symbiodinium* DNA with minimal clade bias in comparison to other commonly used primers. In particular, the SYM_VAR primer pair's ability to amplify from *Symbiodinium*-rare environments whilst maintaining maximum taxonomic resolving power, holds potential for better elucidating the role of free-living *Symbiodinium* in reef ecology. Our standardised protocol should therefore help maximise return of information from a diverse range of sample types whilst maintaining comparability to future and existing projects.

## ACKNOWLEDGEMENTS

*Tara* Pacific consortium acknowledgement: We are keen to thank the commitment of the people and the following institutions and sponsors who made this singular expedition possible: CNRS, CSM, PSL, KAUST, Genoscope/CEA, ANR-CORALGENE, France Genomique (specifically grant number ANR-10-INBS-09), agnès b., the Veolia Environment Foundation, Region Bretagne, Serge Ferrari, Billerudkorsnas, AmerisourceBergen Company, Lorient Agglomeration, Oceans by Disney, the Prince Albert II de Monaco Foundation, L'Oréal, Biotherm, France Collectivités, Kankyo Station, Fonds Français pour l'Environnement Mondial (FFEM), Etienne Bourgois, UNESCO-IOC, the *Tara* Foundation teams and crew. *Tara* Pacific would not exist without the continuous support of the participating institutes. This work is contribution number 4 of *Tara* Pacific.

### Funding

This project has been funded through the *Tara* consortium, France Genomique grant number ANR-10-INBS-09, and KAUST baseline and KAUST BESE division funds to Christian R. Voolstra. The funders had no role in study design, data collection and analysis, decision to publish, or preparation of the manuscript.

### Grant Disclosures

The following grant information was disclosed by the authors:
*Tara* consortium, France Genomique: ANR-10-INBS-09.
KAUST baseline and KAUST BESE division funds.

### Competing Interests

Xavier Pochon is an Academic Editor for PeerJ.

## Author Contributions

- Benjamin C.C. Hume conceived and designed the experiments, performed the experiments, analysed the data, prepared figures and/or tables, authored or reviewed drafts of the paper, approved the final draft.
- Maren Ziegler conceived and designed the experiments, performed the experiments, analysed the data, authored or reviewed drafts of the paper, approved the final draft.
- Julie Poulain conceived and designed the experiments, performed the experiments, analysed the data, authored or reviewed drafts of the paper, approved the final draft.
- Xavier Pochon performed the experiments, analysed the data, authored or reviewed drafts of the paper, approved the final draft.
- Sarah Romac performed the experiments, contributed reagents/materials/analysis tools, authored or reviewed drafts of the paper, approved the final draft.
- Emilie Boissin performed the experiments, contributed reagents/materials/analysis tools, authored or reviewed drafts of the paper, approved the final draft.
- Colomban de Vargas performed the experiments, contributed reagents/materials/analysis tools, authored or reviewed drafts of the paper, approved the final draft.
- Serge Planes performed the experiments, contributed reagents/materials/analysis tools, authored or reviewed drafts of the paper, approved the final draft.
- Patrick Wincker conceived and designed the experiments, performed the experiments, contributed reagents/materials/analysis tools, authored or reviewed drafts of the paper, approved the final draft.
- Christian R. Voolstra conceived and designed the experiments, performed the experiments, analysed the data, contributed reagents/materials/analysis tools, prepared figures and/or tables, authored or reviewed drafts of the paper, approved the final draft.

## Field Study Permissions

The following information was supplied relating to field study approvals (i.e. approving body and any reference numbers):

The following permits were acquired for the collection of the samples analysed as part of this study: Panama scientific sampling: SE/AP-18-16; Panama CITES: SEX/A-72-16, no. 05567; Portugal scientific sampling: 080522/2009/Proc.o E.17.b; Maldives scientific sampling: FA-D2/33/2010/02; Chile scientific sampling: 13270/24/74.

## Data Availability

Sequence data can be accessed through Bioproject accession number PRJNA430028.

## Supplemental Information

Supplemental information for this article can be found online at http://dx.doi.org/10.7717/peerj.4816#supplemental-information.

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
