# Peer review of "An improved primer set and amplification protocol with increased specificity and sensitivity targeting the Symbiodinium ITS2 region"

_PeerJ, doi:10.7717/peerj.4816_

## Round 0.1 · original submission · Major Revisions

Dear Authors,

I have received two reviews, one minor revision and one rejection. I also reviewed the MS myself. The minor revision brings up substantial points and issues, and at the same time your MS does not warrant a rejection on the grounds pointed out by the second reviewer. Thus I am recommending a major revision.

In your revision I would like to address these specific points:

1. Your protocol amplified all 9 clades of Symbiodinium consistently at a give temperature. It also amplifies all Symbiodinium clades associated with highly divergent coral groups consistently. So I am assuming that it will amplify consistently Symbiodinium clades consistently with all the different species of coral that are part of the clade represented by your samples. But in spite of this, it would like an explicit statement of which species were used in your analyses, and if this protocol was applied to other species (e.g. range of species to which your protocol was applied in your lab).

2. Different PCR machines, ramping times, etc. probably fall into the same category as different polymerase systems, that is there might be quantitative differences but not qualitative differences. I would appreciate if you could elaborate on this.

3. Make a detailed protocol available publicly, e.g. lab website. This will increase the utility of this protocol.

4. Finally the MS still needs editing; sentence structure and composition is at time awkward.

Otherwise I believe this is a fine contribution to the community, and a step towards the standardization of protocols that will result in higher quality and greater consistency data that will be comparable across studies.

Sincerely,

Tomas Hrbek

Reviewer 1 ·

Basic reporting

The manuscript is well written and follows a construction that adheres to the typical formatting guidelines required of original research. The authors provide a fair representation of the state of the field, however, some important references are missing in support of statements, and these are detailed in the relevant sections below. The samples analysed in this study are part of a larger set of samples collected in the TARA Oceans/Pacific voyages, which have committed to providing the RAW sequenced data. This is detailed in the manuscript. While it is clear that this manuscript is a precursor to forthcoming analyses of much-larger datasets, the relatively simple set of experiments represent a stand-alone manuscript and not the inappropriate subdividing of data.

Experimental design

The aims and objectives are clearly stated and the study addresses a knowledge gap that will be of interest to many coral/Symbiodinium biologists. The methods are generally well-described with sufficient detail if to be repeated. However, there are some concerns as to the factorial design regarding the optimisation of the SYM_VAR primer set and that what was carried out does not match the stated objectives. One of the objectives was to test the number of cycles on the detection of Symbiodinium in samples where cells may be naturally low in abundance (e.g. environmental samples such as open ocean water). Line 255 – “…and whether an increase in the number of PCR cycles actually aids in the amplification of Symbiodinium from samples representing Symbiodinium-rare environments”. However, the opposite was performed during optimisation. Cycle number was actually decreased from 35 cycles, which is the standard protocol for all primer pairs outlined in table 1, to 30 or 25 cycles depending on the treatment. This does not accomplish the objective as directly stated above and is somewhat perplexing. If you have a more specific primer set, why wouldnt you take your cycle number from 35 to 40, or higher, to better detect rare sequences? This objective needs to be either reworded or removed and the results/discussion altered accordingly. Further complications include a optimisation design that is not fully factorial. For example, you find minor differences in the chimeric reads across the two cycle-number treatments, but could this difference be due to the change in annealing temp to 59 C? Why was a 59 C treatment with 35 cycles not conducted? Again, the objective likely needs to be reworded to reflect what was actually conducted. I suspect this lack of a fully-factorial design is also what precluded the use of simple, but informative, statistics that validate the differences seen across the different optimising treatments. While these are relatively straightforward optimisation steps that other researchers could also conduct, one expected outcome of this manuscript is for labs/groups to decide whether or not to adopt this as their new standard protocol, and this ambiguity makes the choice substantially more difficult.

Validity of the findings

The finding that the SYM_VAR primer set is significantly more specific for Symbiodinium is quite remarkable and is the main finding of the manuscript. An extra 20 to 30% of the sequences being assigned to your target organism is a non-trivial improvement that will appeal to many people in this field, especially as the costs associated to NGS (while dropping) are still relatively high and thus minimizing non-specific amplification will likely be prioritised accordingly. While the post-QC number of sequences are in many samples comparable between primer pairs, please present (in the text) any differences in the quality of the sequencing between primer pairs (check for any patterns in depth vs number of reads that pass QC).
Furthermore, whilst this information is likely contained in previous publications (Hume et al), it should be briefly discussed HOW this primer pair has achieved its superiority. I.e. differences in the region of binding, the degenerate design, etc, are currently missing from this manuscript.

Additional comments

Minor comments:

142 –Is there already a DOI or deposit number for the Pangea database that can be referenced here?
161 – Check the reference formatting and spelling of author names.
203 – Change spelling to Mothur
236 – Why this e-value? This seems very high. First interpretation of this value is that we would expect 50 matches to occur simply by chance? Please clarify the use of this threshold.
260 – Which manufacturer? Of the PCR reagents?
275 – Check spelling. “cycle number, OR annealing temperature”.
281 – Please provide some coefficient of variation for these % values here, and throughout the results section.
286 – Does this “respectively” refer to the previous paragraph?
381 – For samples where you only recover 3 out of the 9 known clades, which is a low diversity compared to what is recently being described from reef sediments (7+ clades, Quigley et al 2017 Frontiers), there are only limited conclusions you can draw related to intra-cladal bias. You do suggest the conservative approach, that 56 C is the better option given the small potential for bias, but it should be discussed how further optimisation and assessment of intra-cladal bias need to be carried out on either artificial communities or natural communities with a greater intra-cladal diversity.
387 – This is not what you stated you would do. Above you state you will increase the number of cycles.
417,18 – This statement requires some supporting citation. As you mention, formal species descriptions thus far make limited use of the ITS2 marker or of its intragenomic variants. It is not clear for which clades or sub-clades ITS2 represents a true super-species, species, or sub-species level marker. At this stage species level descriptions have not developed beyond the requiring of other makers in support of the ITS2 data, including 28s, cp23s, and psbAncr.
Figure 1 and 4 – While the data within each panel are clear, they are displayed with an inefficient use of space and could be reformatted accordingly.
Table 2 – There appears to be an extra annealing temp value in line with ITS-DINO/ITS2Rev2
Table 3 – Please increase the decimal places on the chim/total column (or remove this column) as you already have the chimeric seq number in the table and some are clearly non-zero.

Reviewer 2 ·

Basic reporting

The authors explained clearly the purpose of this paper and the language used was clear. The structure of the article generally speaking is professionally made. A few references are missing but this is a very easy fix. I have added comments on the pdf file.

Experimental design

The research question is very relevant as the question of which primers perform the best under the conditions that the authors tested would attract attention from the increasing number of people assessing Symbiodinium diversity. However, I have explained in the test that I have detected some serious flaws in the experimental design; namely the missing info on the species names and the very low number of replications per potential species. The genomic part is commendable except that some details are missing such as the PCR machine used for the initial amplifications.

Validity of the findings

I have explained on the text that lack of basic information regarding the taxonomic ID of the corals and hydrocoral species and the very low number of individuals per species is limiting the potential value of this contribution.

Annotated reviews are not available for download in order to protect the identity of reviewers who chose to remain anonymous.

---

## Round 0.2 · accepted · Accept

Dear Authors,

Your manuscript has been re-reviewed by one of the original reviewers and myself. We both agree that you have addressed the issues brought up in the first review satisfactorily, and that the MS is now acceptable for publication.

There are a couple of minor editorial issues in your MS that I corrected using the track-changes option. I will be sending this to you separately. Please incorporate these corrections, and then you can send the final version of the MS to the editorial office.

Congratulations on a job well done, and a fine contribution to the community.

Sincerely,

Tomas Hrbek

# Reviewer 1 ·

Basic reporting

The authors have sufficiently addressed my concerns with the previous version of the manuscript and I recommend that it is accepted for publication.

Experimental design

The authors have sufficiently addressed my concerns with the previous version of the manuscript and I recommend that it is accepted for publication.

Validity of the findings

The authors have sufficiently addressed my concerns with the previous version of the manuscript and I recommend that it is accepted for publication.